# The Application of Single-Cell RNA Sequencing in the Inflammatory Tumor Microenvironment

**DOI:** 10.3390/biom13020344

**Published:** 2023-02-10

**Authors:** Jiayi Zhao, Yiwei Shi, Guangwen Cao

**Affiliations:** 1Department of Epidemiology, Second Military Medical University, Shanghai 200433, China; 2Shanghai Key Laboratory of Medical Bioprotection, Shanghai 200433, China; 3Key Laboratory of Biological Defense, Ministry of Education, Shanghai 200433, China

**Keywords:** scRNA-seq, immunology, inflammatory tumor microenvironment

## Abstract

The initiation and progression of tumors are complex. The cancer evolution-development hypothesis holds that the dysregulation of immune balance is caused by the synergistic effect of immune genetic factors and environmental factors that stimulate and maintain non-resolving inflammation. Throughout the cancer development process, this inflammation creates a microenvironment for the evolution and development of cancer. Research on the inflammatory tumor microenvironment (TME) explains the initiation and progression of cancer and guides anti-cancer immunotherapy. Single-cell RNA sequencing (scRNA-seq) can detect the transcription levels of cells at the single-cell resolution level, reveal the heterogeneity and evolutionary trajectory of infiltrated immune cells and cancer cells, and provide insight into the composition and function of each cell group in the inflammatory TME. This paper summarizes the application of scRNA-seq in inflammatory TME.

## 1. Introduction

In recent decades, research on the tumor microenvironment (TME) has changed the traditional view of tumor treatment centered on tumor cells. The initiation and development of cancer depend on the genetic heterogeneity of malignant cells and the TME. The inflammatory TME plays a pivotal role in cancer development, affecting cancers at various stages, including initiation, promotion, malignant conversion, invasion, and metastasis [1]. Cancer evolution-development (Cancer Evo-Dev) is a theory of inflammation-induced oncogenesis [2]. The components of this theory are as follows: the dysregulation of immune balance induced by inherited and acquired environmental factors maintains non-resolving inflammation, which promotes tumorigenesis and progression, and proceeds to cancer development with the characteristic of “variation-selection-adaptation.” In the context of the non-resolving inflammatory TME, pro-inflammatory factors promote somatic mutations and viral mutations by either trans-activating the expression of nucleic acid editing enzymes and their analogs or by inducing oxygenation excitation. The inflammatory TME is an essential component of all tumors, even tumors in which a direct causal relationship with inflammation has not yet been demonstrated [3]. Immune and inflammatory responses play several roles at various stages of tumor development, and the balance between immunity and inflammation is crucial to malignancy development and progression [4]. Therefore, it is essential to study inflammatory TME and elucidate its role in tumor initiation and development; doing so may provide insight into developing effective cancer therapy.

Single-cell RNA sequencing (scRNA-seq), a next-generation sequencing technique, has revolutionized our ability to study the TME and improve the progression of tumor research. scRNA-seq measures cell transcription at the single-cell resolution level, providing an unprecedented view of the complexity of the TME [5]. The advantages of scRNA-seq inspire researchers to apply this technology and elucidate the mechanisms by which subsets of cells in TME promote cancer evolution and heterogenicity. This review provides an overview of scRNA-seq and its application in immunology and inflammatory TME.

## 2. Overview of scRNA-seq

Until recently, investigators performed embryology, immunology, physiology, and oncology studies using traditional transcriptome sequencing. This bulk sequencing averages the expression levels of RNA, ignoring the heterogeneity of gene expression among cells within a population; as a result, some valuable information may be missed [6]. The emergence of scRNA-seq solves this problem by describing the RNA expression profile of each cell. With bioinformatics, scRNA-seq distinguishes the differences in gene expression between cells and identifies rare cells in heterogeneous populations [7,8].

scRNA-seq refers to RNA sequencing that can detect transcription at the single-cell resolution level. This technology can be divided into several steps: organ dissociation, single-cell capture, cell lysis, mRNA reverse transcription, cDNA amplification, library construction, high-throughput sequencing, and data analysis [9]. Current single-cell RNA sequencing methods include Tang methods, CEL-seq, SMART-seq, STRT-seq, and Drop-seq [10]. Table 1 compares these five methods.

Single-cell capture determines the cost and scale of single-cell sequencing. The mainstream single-cell capture technologies include microfluidic platforms, such as Dropseq, Chromium, and InDrop, which improve the mRNA capture efficiency [16,17]. In the water-in-oil environment, single cells are combined with gel beads containing specific oligonucleotide tags (barcodes), and different specific sequences are attached to each cell. Then the droplets are mixed, the cells are lysed, and reverse transcription is performed (Figure 1). This method is available for low-cost, high-throughput single-cell sequencing [18]. In addition to the microfluidic platform, single-cell capture includes micromanipulation [19] and laser capture microdissection [20].

The efficiency of mRNA reverse transcription is critical for determining the sensitivity and accuracy of scRNA-seq. There are three mainstream methods for reverse transcription: (1) poly(A) tailing followed by PCR, as in Tang-seq [11]; (2) second-strand synthesis followed by in vitro transcription (IVT), such as CEL-seq2 [21] and MARS-seq [22]; and (3) a template-switching method, as in STRT-seq11 [14] and Smart-seq/Smart-seq2 [23]. The third method is the most popular because it reduces the 3′ coverage deviation caused by incomplete reverse transcription, obtains complete transcript coverage, and requires fewer reaction steps [24].

When performing scRNA-seq analysis, the optimal sequencing method should be adopted according to the experimental purpose. In the subsequent steps of library construction and data analysis, scRNA-seq and traditional bulk sequencing are based on the same principles and processes.

Previous bulk cell sequencing technologies analyzed cells at the population level, and the average data obtained usually reflected the dominant cell data, which often ignored the differences between individual cells and masked the characteristics of rare samples. In recent years, scRNA-seq has begun to provide insight into the study of cell heterogeneity in multicellular organisms. It demonstrates changes in each cell by analyzing the expression profile of the cell transcriptome at the single-cell level and discovering new cell types. For these reasons, scRNA-seq has been used in oncology [25,26], immunology [6], virology [27], embryology [28], and microbiology [29].

## 3. scRNA-seq in Immunology

The immune system comprises many immune cells, including innate and adaptive immune cells [30]. These cells are critical for protecting the host from injuries, infection, and carcinogenesis (i.e., immune surveillance). Improvements in technologies such as microscopy and flow cytometry have accelerated the classification of immune cells [31]. Nevertheless, these methods still have some limitations [6]. The immune system’s complexity, including the heterogeneity, development, differentiation, and microenvironment of immune cells in health and disease, cannot be fully understood using classical theories [32]. A deeper inspection of immunology yields improved immune therapies using advanced technologies. The emergence of scRNA-seq can revolutionize our understanding of immunology and break through the bottlenecks in immunology. The following section illustrates the application of scRNA-seq in immunology, including identifying novel immunocyte subsets and novel marker genes, revealing heterogeneity, reconstructing the trajectory of immune cell development and differentiation, and uncovering immune mechanisms.

### 3.1. Identification of Novel Immunocyte Subsets and Marker Genes

Immune cells have many functions, including killing infected and mutated cells, priming the adaptive immune response, and causing chronic inflammation [33]. However, immune cells’ functions are complicated. First, the same immune cells may perform different functions, sometimes even opposite functions. Second, the same T cell subset may serve dual roles. What causes the same immune cells/subsets to serve different or opposite functions remains unclear, hindering the development of successful immunotherapy. One possibility is that several distinct cell subsets with different phenotypes and functions coexist in the TME. Previously, the identification of immune cells depended on our knowledge of surface markers. However, scRNA-seq can define cell types using transcriptome analysis, which enables the identification of novel cell subsets and their marker genes using DEGman [34]. Data analysis from scRNA-seq using novel bioinformatics methods might accurately identify this heterogeneity and new subsets of immune cells.

Reliable markers are critical for the early diagnosis of some diseases and can lead to better treatments and outcomes. Complement-secreting CAFs (csCAFs) were discovered using scRNA-seq. In addition, several new markers of regulatory and exhausted T cells, such as DUSP4, FANK1, LAIR2, and DUSP4, are new signature genes for regulatory T cells (Treg), which can serve as prognosis markers and potential therapeutic targets [5]. scRNA-seq has been employed to compare the transcriptome data between primary tumors and their metastatic counterparts in pancreatic neuroendocrine tumors. A novel gene signature (PCSK1 and SMOC1) defining the metastatic potential of the tumor and its prognostic value has been identified and validated [35]. These studies demonstrated that scRNA-seq could identify novel immunocyte cell populations, subpopulations, and novel marker genes; it could identify disease-specific cells and act as prognosis markers.

### 3.2. Revealing the Heterogeneity

The heterogeneity in immunity has attracted substantial attention [36,37,38,39,40]. The immune system is a complex network in which the types, states, and locations of immune cells are diverse in healthy individuals and patients [32]. Active research areas include the heterogeneity of immune systems involving immune cell heterogeneity, immune responses, and the immune microenvironment. scRNA-seq can sequence the smallest independent genetic units in life to address problems derived from heterogeneity that cannot be resolved using bulk RNA sequencing [41,42]. scRNA-seq holds substantial potential for revealing immunological mechanisms regarding tissue composition, transcription dynamics, the regulatory relationships between immune genes, and the heterogeneity of the traditional “same” immune cell types.

When fighting pathogens and diseases, immune cells are highly heterogeneous [43]. Heterogeneity is usually defined by surface markers that can be measured using flow cytometry or the giant cell technique. However, these methods are limited by the number of surface markers. A more robust method of cell typing is based on scRNA-seq, which facilitates elucidating heterogeneity without understanding gene functions. Although PD-LI/PD-1 can be treated as promising targets for immunotherapy and have been applied to cancers such as lung, breast, and hematologic cancers, they are ineffective for other cancers. The reason may be the intratumoral heterogeneity of immune cells, which complicates the identification of effective immunotherapeutic targets. By applying scRNA-seq to analyze the heterogeneity of immune cells in ovarian cancer, researchers demonstrated that tumor-infiltrated myeloid cells are heterogenous across patients [44]. Thus, scRNA-seq should help develop specific immunotherapy for cancers by analyzing immune cell heterogeneity.

The response may differ by using the same immunotherapy to treat patients with the same malignant disease because immune response heterogeneity is common. The reason behind this may be related to the heterogeneity of the TME. scRNA-seq has been used to investigate whether the pancreatic ductal adenocarcinoma (PDAC) response to an immune checkpoint inhibitor is enhanced by changing the TME via targeting CD47 with a monoclonal antibody. Various PDAC mouse models displayed varying responses to anti-CD47 and anti-PD-L1 blockade because the TME established by different PDAC cell lines varied across these models. These findings suggest that targeting CD47 may alter the TME and alter the response of PDAC to immune checkpoint inhibitors [45].

Tumor metabolism is dynamic, and the components of TME change constantly. A study analyzed the dynamics of the transcriptome profile of immune cells at the single cell level from the primary state to metastasis of a pancreatic neuroendocrine tumor and found that the immune microenvironment of the primary lesions was distinct from the metastatic ones, revealing the intra- and inter-humoral heterogeneity of cell populations [35]. Because the immune system is a complex network for defending against various diseases, heterogeneity in immune cells and immune responses is common. The heterogeneity of immune cells also contributes to immune tolerance [46]. The heterogeneity and dynamics of cellular components in the TME at various stages of cancer development should be characterized before performing immunotherapy.

### 3.3. Reconstructing the Trajectory of Immune Cell Development and Differentiation

scRNA-seq can also be used to detect immune cell development, including differentiation [47,48,49], maturation [50], responses to stimulation, and activation [51]. Reconstructing the trajectory of immune cell development and differentiation at the single cell level can provide valuable insight into the immune microenvironment and locate immune cells’ origin and critical events in disease progression [52]. The immune microenvironment is complicated and dynamic. Understanding the dynamics of immune/inflammatory cells can deepen our insight into TME.

By comparing samples with different stages of PDAC based on scRNA-seq, the proportions of classical CAFs (cCAF), csCAFs, and pancreatic stellate cells (PSCs) experience significant changes from early PDAC to late PDAC. Investigation into the developmental trajectories of these subpopulations demonstrated that csCAFs suppress tumors in the TME of PDAC and decrease during progression; the TME of late PDAC contains only PSCs but not cCAFs or csCAFs. The evolution of cCAFs and csCAFs towards PSCs may be a strategy to convert anti-tumor CAFs into pro-tumor CAFs [5]. During PDAC progression, an anti-tumor immune response occurs but is disabled by negative regulation from accumulated Tregs, exhausted T cells, and tumor-associated macrophages (TAMs).

Trajectory analysis helps investigate the origins of immune cells and neoplastic cells. An analysis was performed between two subgroups of ductal cells and acinar cells, revealing that acinar cells could become type 1 ductal cells and then transform into type 2 ductal cells in PDAC [53]. Another study described the evolutionary trajectory of immune cells using scRNA-seq, and critical molecular events were identified in the progression from primary lesions to metastases [35]. Metabolic reprogramming develops at the “mid-late” stage, supporting the notion that tumor metabolism is a dynamic process and is adapted to the TME. scRNA-seq was used to study bladder urothelial carcinoma and revealed that monocytes undergo M2 polarization in the tumor region and differentiate into TAMs, while the LAMP3-positive dendritic cells (DCs) recruit Tregs. These immune cells with inflammatory CAFs (iCAFs) potentially take part in the formation of an immunosuppressive TME, therefore playing a role in tumor progression [54]. These representative studies provide insights into cancer immunology and provide an essential resource for immunotherapy and drug discovery.

### 3.4. Uncovering Immune Mechanisms

Thanks to the work of the innate immune system and the adaptive immune system, health can be maintained. Chronic inflammation occurs because of abnormalities in the innate immune system, whereas exercise reduces chronic inflammation involving immunology and oxidative stress [55]. A deep understanding of the immune mechanisms of chronic inflammation and immune escape is critical, focused on how the immune mechanism protects the host and on why the immune mechanism loses its ability to protect the host. scRNA-seq can uncover these immune mechanisms. Immune escape (i.e., when the host loses the ability to defend itself against the foreign invaders) leads to disease progression. Immune surveillance evasion is the hallmark of tumor progression [56]. By analyzing the scRNA-seq data from a patient with multidrug-resistant Mantle cell lymphoma, several potential immune escape mechanisms of malignant cells were identified, including the anti-perforin pathway and the low tumor cell immunogenicity pathway [57]. Similarly, scRNA-seq in bladder urothelial carcinoma suggests that bladder cancer cells may evade immune detection by down-regulating immunogenicity [54].

The immune microenvironment is involved in osteosarcoma tumorigenesis and progression. However, the distribution and dynamics of immune cells in osteosarcoma are not well understood. scRNA-seq data in osteosarcoma suggests that a cluster of regulatory DCs forms the immunosuppressive TME by recruiting Treg cells. The major histocompatibility complex class I is downregulated in osteosarcoma [58]. These findings suggest that decreased tumor immunogenicity may be the potential mechanism of immune escape.

Lineage plasticity and stemness contribute to drug resistance in cancer therapy because these flexible states allow cancer cells to dedifferentiate into stem-like cells. Understanding how the immune system works in drug resistance may provide insights into immune therapy against cancer. A study reported scRNA-seq in malignant and microenvironment cells in patients with relapsed/refractory early T-cell progenitor acute lymphoblastic leukemia carrying activating NOTCH1 mutations; the investigators detected two functionally different stem-like states, irrespective of cell cycle or oncogenic signaling. More importantly, these two stem-like states differentiated into mature leukemia states, suggesting that they have prominent immunomodulatory functions, promoting an immunosuppressive leukemia ecosystem with the clonal accumulation of dysfunctional CD8+ T cells that expressed HAVC [59]. This finding suggests therapeutic targets based on cellular states might limit cancer cells’ molecular escape.

Colorectal cancer (CRC) is a typical “cold” cancer characterized by insufficient immune cell infiltration. Overcoming the deficiencies in CRC treatment might depend on improving our understanding of TME. scRNA-seq data from ten human CRC samples from the Gene Expression Omnibus (GEO) database revealed that differentiation is accompanied by remodeling of lipid metabolism and suppression of immune function, suggesting that lipid remodeling may be a fundamental cause of immunosuppression [60]. Thus, scRNA-seq could uncover immune mechanisms, enriching our understanding of cancer immunology and leading to the development of specific cancer immunotherapies.

## 4. scRNA-seq in the Inflammatory TME

Several inflammatory cells infiltrate the TME, including macrophages, granulocytes, monocytes, and mast cells [2,61]. These cells interact with the tumor by secreting cytokines, which constitute a complex inflammatory TME, exhibit an immunosuppressive effect, facilitate the retro-differentiation of mutated cells, and affect the tumor’s biological process. The complexity of the inflammatory TME is reflected in the diversity of the infiltrated inflammatory cells, the secretion of the inflammatory factors, hypoxia, hypoxia-inducible factors, acidity, and low sugar interactions between different components [62], and it is also reflected in the plasticity of inflammatory cell infiltration and cell differentiation. In the TME, the mutated cells are retro-differentiated into tumor-initiating cells or stem cells. The phenotype and function of the somatic cells change, promoting tumor progression. In addition to pro-inflammatory cell infiltration, the TME includes innate immune cells (natural killer cells), adaptive immune cells (Tregs, B cells), cancer cells, CAFs, endothelial cells, and the surrounding matrix. These cells communicate with one another through direct or indirect contact, by secreting cytokines and chemokines, and by controlling and shaping tumor growth through autocrine and paracrine pathways. The expression of various immune mediators and regulators in the TME and the abundance and activation status of different cell types determine the direction of immune balance and whether inflammation promotes or inhibits tumor growth [63]. Therefore, it is essential to characterize the inflammatory infiltrating cells to understand how immune mediators and regulators work and to dissect the inflammatory TME (Figure 2). scRNA-seq is a powerful tool for studying the pro-inflammatory TME because it reveals the functional status of cells at the single-cell level. 

### 4.1. Analyzing the Composition and the Heterogeneity of the Inflammatory TME

The inflammatory TME comprises many highly heterogeneous cell types. Various types of cells play different roles in tumor development, forming a variety of inflammatory TMEs. Because of the heterogeneity of TME, the tumor acquires the capacity to generate drug resistance [10]. The heterogeneity of TME also plays a crucial role in cancer promotion; therefore, an in-depth understanding of the gene expression patterns of a single cell is critical [64,65]. Traditional RNA sequencing can only produce results after mixing a large amount of intracellular gene expression information from multiple species. Traditional RNA-seq does not identify the different cell types or their status in the inflammatory TME; it also does not identify rare cell clones, which may be related to tumor progression [66]. scRNA-seq overcomes this obstacle by identifying different inflammatory cell groups in the TME. It identifies biomarkers that can be used to characterize these cells at the single-cell level, revealing their developmental and functional status and the heterogeneity of inflammatory TME.

Appling scRNA-seq to samples from gastric cancer (GC) and normal tissues, seven major cell types in the TME were identified: epithelial cells, endothelial cells, fibroblasts, T cells, B cells, macrophages, and mast cells. CAFs (an essential part of the TME) play diverse roles in GC promotion, and CAF subtypes were identified in myofibroblasts, pericytes, extracellular matrix CAFs, and iCAFs. ICAFs participate in tumor development and are related to poor prognosis [67].

Similarly, the multiple myeloma inflammatory stromal cell landscape was mapped using scRNA-seq. Specific inflammatory mesenchymal stromal cells (iMSCs), including MSC1 and MSC2, were evident in myeloma samples. IMSCs are spatially colocalized with tumor and immune cells and transcribe genes involved in tumor survival and immune regulation. IMSCs regulate bone marrow cell functions and interact specifically with proliferating myeloma cells, suggesting that iMSCs participate in tumor development and survival [68].

Using scRNA-seq to analyze lung adenocarcinomas harboring EGFR mutations, 12 heterogeneous subclusters of myeloid cells and T cells were identified as the most abundant stromal cell types in the TME. Accompanied by an increase in CD1C+ DCs, the TAMs showed pro-tumoral functions. Tumor-infiltrating T cells displayed exhausted and regulatory T cell features. The adenocarcinoma cells were categorized into subtypes based on their gene expression signatures in distinct pathways, including hypoxia, glycolysis, cell metabolism, translation initiation, cell cycle, and antigen presentation [69]. The intra- and inter-tumor heterogeneity in the TME and tumor cells can be addressed using scRNA-seq. scRNA-seq can generate the transcriptional profiles of these individual cells. Using these data, cluster analysis of these cells can classify them, identify the composition of the inflammatory TME, and reveal the intra- and inter-tumoral heterogeneity.

### 4.2. Describing the Developmental Trajectory of Inflammatory Cells

Various types and states of inflammatory cells are present in the inflammatory TME. Developmental inter-transformation often occurs between different states of cells, leading to immune suppression. scRNA-seq can provide complete and detailed transcriptome data. Pseudotime analysis detects the developmental trajectory of cells through biological processes based on transcriptional similarities. Using algorithms such as Monocle 2 [70] to analyze the scRNA-data, investigators determined where these inflammatory cells come from and where they go, helping to clarify how inflammatory cells function in tumors.

Using pseudotime analysis to analyze scRNA-seq data, mesenchymal stromal cells (MSC) 2 were identified as an intermediate cell in the differentiation process from non-inflammatory MSCs to inflammatory MSCs (iMSCs), suggesting that MSC activation is sufficient to induce the inflammatory transcriptome associated with multiple myeloma-specific iMSCs [68].

Monocle 3 was used to analyze the scRNA data and calculate the pseudo-temporal sequence of monocyte differentiation to determine how inflammatory responses affect myeloid differentiation. This analysis revealed that aging and inflammatory signals significantly altered the developmental trajectory of Tet2^HR^ mutated bone marrow cells, which are imprinted early in the differentiation stage and promoted the emergence of MHC IIhi monocytes. These findings suggest that enhanced inflammatory signaling in TET2 mutant mice leads to inflammation with pathologic potential and initiates the production of abnormal monocyte subsets [71].

To identify strategies to coordinate the immune system and hepatocellular carcinoma (HCC), it is critical to understand the mechanisms and pathways of CD8+ T cell exhaustion and Treg accumulation in tumor tissues. With complete transcriptome data and T-cell receptor information for many T cells, investigators delineated the developmental trajectories of CD8+ T cells. They found that naive CD8+ T cells were initially, followed by effector memory CD8+ T cells and exhausted CD8+ T cells, suggesting the transition of T cell status from activation to exhaustion [72]. Another study investigated the ecosystem in early-relapse HCC using scRNA-seq. Compared with primary tumors, early-relapse HCCs have reduced levels of Treg cells, increased DCs, and increased infiltrated CD8^+^ T cells; however, CD8^+^ T cells in recurrent tumors display an innate-like low cytotoxic state with low clonal expansion, while DCs have dampened antigen presentation [73]. These studies suggest that the TME in HCC tissues recruits innate-like or exhausted CD8^+^ T cells and dampens DC antigen presentation, leading to tumor relapse-related immune evasion.

### 4.3. Predicting the Prognosis of Different Types of Inflammatory TME

Single-cell sequencing generates substantial amounts of information. In addition to specific biological information tools for analysis, cell groups and characteristic genes determined by single-cell sequencing data can be used to analyze existing traditional sequencing databases such as The Cancer Genome Atlas (TCGA) and obtain clinical outcome information corresponding to different inflammatory TMEs.

Using scRNA-seq on tumor samples and para-tumor samples from bladder carcinoma and the sequencing data from TCGA, investigators found that the iCAF-specific marker PDGFRA was associated with poor overall survival while the myo-cancer-associated fibroblast marker RGS5 was not [54].

Single-cell transcriptomic profiling of hematopoietic stem and progenitor cells into differentiated monocytes was analyzed in an animal model carrying a recurrent TET2 missense mutation who developed a wide range of myeloid neoplasms late in life. The MHC IIhi monocyte gene signature was significantly associated with survival in a cohort of young patients with TET2 mutations. MHC IIhi monocyte characteristics are associated with significantly shorter survival, demonstrating that MHC IIhi monocyte has “biomarker potential” [71].

Three subtypes were identified based on the gene expression data of the identified malignant epithelial cells (mEPCs). Based on tissue microarray data from another independent gallbladder carcinoma (GBC) cohort, the authors explored the relationship between mEPC subtype-specific markers and prognosis. Increased expression of MUC2 (a marker for subtype I) was significantly associated with overall survival, while the high expression of CTSD (a marker for subtype II) and MSLN (a marker for subtype III) may predict shorter overall survival [74]. Thus, scRNA-seq, with existing traditional sequencing databases, detects some novel cell subsets and markers and evaluates their prognostic significance.

### 4.4. Revealing the Interaction between Inflammation and Tumors

By comparing a single-cell atlas of chronic inflammation and cancer or comparing tumors at different stages, investigators can characterize the dynamic changes in cell groups in the TME at various stages and observe the dynamic changes of inflammatory cells. More importantly, the interaction between inflammation and tumors can be identified. scRNA-seq can provide a wide-angle perspective to identify niches and networks in a complex cellular matrix and distinguish cell subsets during cancer initiation, progression, and metastasis [75]. This single-cell landscape may lay the foundation for solving the mystery of cancer initiation and metastasis, revealing targets for chemotherapy and immunotherapy.

Premalignant lesions, including chronic atrophic gastritis and intestinal metaplasia, precede intestinal-type GC. Using scRNA-seq, investigators found that mucous gland cells tended to acquire an intestinal-like stem cell phenotype during metaplasia. OR51E1 is a marker for individual endocrine cells in the early-malignant lesion, while HES6 identifies pre-goblet cell clusters, potentially aiding the identification of metaplasia at the early stage [76].

By mapping representative genes and cytokines in the NF-κB signal pathway to these cell types, the cell origin of mediators in the cytokine and NF-κB signal pathways were mapped and found to be associated with gastritis-induced GC. This finding suggests that the TME orchestrated by inflammatory cells is indispensable in the tumor process [77]. Macrophages may be the primary source of interleukin 1β and PTGS2; this genotype has a greater GC risk, and PTGS2 plays a critical role in mediating inflammation by activating NF-kB [78], suggesting that macrophages in the inflammatory microenvironment promote GC development.

The proportional changes of inflammatory cells in TME can help elucidate the mechanism of tumor progression. To describe the transcriptional events that lead to acute myeloid leukemia progression in Tet2^HR^ mutant animal models, scRNA-seq was performed on total bone marrow cells collected from young animals with no signs of disease and older animals showing chronic (MDS/MPN-like) and transforming myeloid disease (acute myeloid leukemia-like). The loss of TET2 catalytic activity in older animals resulted in an increased percentage of monocytes and a decreased percentage of granulocytes and B cells, suggesting that the acquisition of aging-mediated abnormal transcription programs promotes the transformation of TET2 mutation-associated myeloid tumors. The accumulation of IFN-γ, IL6, IL1B, TNF-α, and IL12 also occurred, suggesting the establishment of a pro-inflammatory microenvironment in the bone marrow of these animals. These findings suggest that the progression of myeloid transformation is associated with increased signals of interventricular inflammatory responses in the bone marrow of diseased animals [71].

To study the dynamic changes in cell compositions and characteristics during inflammation, cancer metastasis, and how inflammation participates in chronic cholecystitis—primary GBCs—metastatic GBCs, samples from these patients were subjected to scRNA-seq. Compared with the single-cell transcriptome atlas from chronic cholecystitis samples, the primary GBC samples contained more endothelial progenitor cells and mesenchymal cells but fewer immune cells, especially T cells as tumor-infiltrating lymphocytes [74]. These differences suggest dynamic adaptation, spatial competition, and the survival of cells in different ecosystems. Within the TME, cells experience competition with their neighbors, with those less fit being eliminated by fitter adjacent cells. The TME may select somatic mutations in cancer cells for their ability to exploit cell competition to kill neighboring host cells, facilitating tumor expansion [79]. By releasing inflammatory mediators, tumors form dense barriers that exclude T cell infiltration, helping construct the immunosuppressive TME.

In recent decades, tumor treatment based on the relationship between TME and the malignant phenotype has emerged as a promising therapy. The interaction between tumor cells and the TME provides a new direction for targeted therapy [80]. scRNA seq can measure cell-cell communication and analyze the relationship between different individual cells to suggest interactions between inflammatory cells, CAFs, and tumor cells in the TME. The scRNA-seq data from CAFs in GC identified four subsets, two of which can communicate with adjacent immune cell subsets in the TME: iCAFs attract and regulate the function of T cells by secreting IL-6 and the C-X-C motif chemokine ligand 12; extracellular matrix CAFs interact with M2 macrophages by expressing periosteal hormone. These subsets show enhanced pro-invasion activity and mobilize surrounding immune cells to build a favorable microenvironment for the tumor [67].

The AT-rich interaction domain 1A (ARID1A) mutation is enriched in hypermutated-single nucleotide variants and microsatellite-unstable subtype GC and predicts responsiveness to fluorouracil-based chemotherapy and PD-1 blockade. ARID1A mutations correlate with immunogenic TME characterized by elevated activated subsets of CD8+ T cells, CD4+ T cells, and natural killer cells. Thus, ARID1A-mutant GC displays immunogenic TME, which might be suitable for combined treatment with chemotherapy and PD-1 blockade [81]. scRNA-seq can provide valuable insight into the interaction among inflammatory cells, CAFs, and tumor cells in the TME.

## 5. Spatial Transcriptomics Combined with scRNA-seq in Tumor, Inflammation, and Immunity

scRNA-seq has augmented the ability to recognize and characterize cell subsets, and transcriptome information can be understood from another dimension [82]. scRNA-seq provides insights into the identification of tumor heterogeneity and cell subsets. However, this technique requires tissue decomposition during sample processing, resulting in the loss of original spatial location information. Spatial location information indicates possible cell interaction, which closely correlates with physiological and pathological functions [83]. The Lundeberg group first reported the concept of spatial transcriptomics, the first space-transfer omics technique based on original capture RNA [84]. Several techniques capable of high-throughput in situ RNA detection and analysis have been grouped under “spatial transcriptomics.” Although the principles of these technologies differ, they share the common feature of recording the spatial location information of the detected RNA. Spatial transcriptomics has been used to study various human diseases [85,86,87]. A significant limitation of spatial transcriptomics is its lack of cell resolution; depending on the tissue, 10-200 cells of the transcriptome are captured at each spot [88]. Because scRNA-seq and spatial transcriptomics have limitations, their combination can provide a comprehensive and unbiased analysis. The following section overviews spatial transcriptomics and its applications in combination with scRNA-seq.

### 5.1. Spatial Transcriptomics

Spatial transcriptomics combines gene expression with an immunohistochemical image of a tissue section, allowing gene expression information from various cells in the tissue to be located in its original spatial location. Genes actively transcribed in the tissue can be distinguished to detect gene expression differences in various tissue parts visually. Spatial transcriptomics reveals the spatial distribution of cell populations and local networks of intercellular communication [89]. Spatial transcriptomics techniques differ substantially concerning the number of detectable genes and the size of detectable tissues, which can be divided into two categories. One is based on in situ capture and sequencing. Before sequencing, the spatial location information is encoded on the transcript using in situ capture, and the transcript and its spatial location information are obtained by high-throughput sequencing.

The second technique is the image-based method. The transcripts are dyed and coded at the original position to achieve multiple detections. The tissue sections are fixed on the poly T-base primer microarray chip with the spatial location information label sequence. The tissues are stained with hematoxylin-eosin and photographed to record the morphology. Then, the tissue undergoes permeabilization and is in situ reverse-transcribed to obtain the mutually complementary DNA with spatial location information. Finally, the gene sequence is obtained using second-generation sequencing, and the spatial location information tag is decoded. The spatial transcriptome data can be obtained through joint analysis with tissue morphology.

Several high-throughput in situ RNA detection techniques have been developed, including Slide-seq [90], high-definition spatial transcriptomics [91], Seq-scope [92], and Stereo-seq [93].

Image-based spatial transcriptomics is performed as follows: After the signal amplification of the original position of the same RNA, different labeling probes are used for multiple label imaging, and the advanced graphical analysis algorithm is used to register and label the colors of each round to obtain a string of color-coded labels. This procedure detects different genes using different color sequences. Thus, highly multiple RNA in situ detections can be achieved without limiting the number of spectral and microscopic detection channels [94].

“Spatial transcriptomics” refers to a class of novel transcriptome analysis techniques that achieve highly multiple RNA in situ detection. They can provide more supporting information for accurately predicting gene function and regulatory networks by linking gene expression information with histomorphological information. However, these technologies often have different detection fluxes and resolutions, and researchers must make reasonable choices according to the target.

### 5.2. Combined Applications in Tumor, Inflammation, and Immunity

Spatial transcriptomics can generate transcriptomic data from a complete tissue section and locate and distinguish the expression of functional genes at specific spatial locations based on scRNA-seq; these procedures allow single-cell transcriptome data to be spatially mapped on high-resolution histopathological images [95]. The combined application of scRNA-seq and spatial transcriptomics can deepen our understanding of cell interactions in the TME.

The combined application can reveal the spatial landscape of diverse cells in the TME. First, scRNA-seq is applied to identify cell populations, then spatial transcriptomics is applied to generate unbiased transcriptomic maps of the tissue sections, and lastly, the data from scRNA-seq and spatial transcriptomics are integrated using multimodal intersection analysis. The results offer the spatial location of each cell population identified, helping detect the cell types enriched most in the tumor regions. For example, by conducting these technologies on esophageal squamous cell carcinoma, stromal cells were more highly enriched in the TME, suggesting that they are involved in tumor initiation and metastasis [96].

The combined application can reveal the interactions inherent in complex tissues. Investigators used these technologies to reveal tissue architecture in PDAC. The subgroups of ductal cells, macrophages, dendritic cells, and cancer cells were limited in their spatial enrichment and co-enriched with other cell types. They also found the co-localization of inflammatory fibroblasts and cancer cells expressing stress response gene modules [97].

The combined application also clarifies the potential origin and regulation mechanisms of abundant cell types in the TME. CRC is a common malignant tumor with limited treatment options. TME analysis can identify potential therapeutic targets. scRNA-seq of samples of CRC demonstrated that tumor-specific FAP^+^ fibroblasts are positively correlated with SPP1+ macrophages, and their tight localization is confirmed by spatial transcriptomics [98]. This interaction may be affected by chemokines, TGF-β, and IL-1, stimulating the formation of immune-excluded desmoplasic structure and limiting the infiltration of T cells. Patients with high expression of FAP or SPP1 enjoy fewer therapeutic benefits than the anti-PD-L1 treatment cohort. These studies provide a potential therapeutic strategy to improve immunotherapy by disrupting the interaction between FAP+ fibroblasts and SPP1+ macrophages. In short, scRNA-seq and spatial transcriptomics offer an unprecedented opportunity to refine the foundation of numerous molecular studies, significantly advancing the development of oncology research and providing guidance for future clinical research and personalized treatment.

## 6. Conclusions and Outlook

After several years of development, scRNA-seq has made significant breakthroughs in technical advancement and clinical applications. This powerful transcriptome-sequencing tool has had a substantial impact on TME research. It made significant contributions to the field, including characterizing the composition of inflammatory TME, describing the developmental trajectories of inflammatory cells, predicting the prognosis of different types of inflammatory TME, and revealing the interaction between inflammation and tumors (Table 2).

Although scRNA-seq has promoted research on inflammatory TME, the application of this technology still has some limitations.

First, because of the rapid development of technology, researchers can obtain substantial information from scRNA-seq; nevertheless, effectively interpreting and using this information presents critical problems. One solution is to combine the results obtained by scRNA-seq (such as some characteristic genes) with existing databases such as TCGA for data mining. In addition, new bioinformatics methods should be developed for the further interpretation of scRNA-seq [99,100].

Second, most of the results obtained by scRNA-seq are descriptive, and many of the data are highly patient-specific. Obtaining universal conclusions with clinical transformation and application requires clarification and discussion. In this regard, researchers should communicate with clinicians to select sequencing samples more precisely according to specific clinical problems.

Finally, during tumor progression, the replication, transcription, and translation of genetic information undergo considerable changes; however, scRNA-seq can only obtain information at the single-cell transcriptome level. Single-cell multi-omics sequencing technology is becoming more prevalent, and the integration of scRNA-seq and other sequencing technology has been widely used. scRNA-seq combined with spatial transcriptomics can reveal the cell interactions in the TME. scRNA-seq will be combined with single-cell genomics, spatial transcriptomics, proteomics, epigenetics, and other sequencing techniques to assist the study of the inflammatory TME.

## Figures and Tables

**Figure 1 biomolecules-13-00344-f001:**
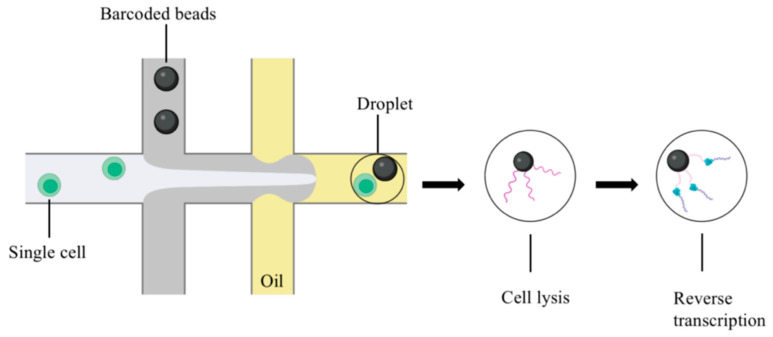
Microdroplet-based scRNA-seq (figure was created with biorender.com accessed on 25 November 2022).

**Figure 2 biomolecules-13-00344-f002:**
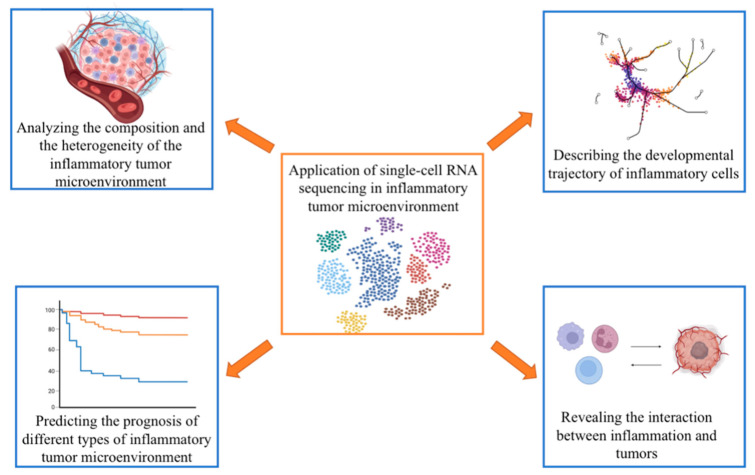
Application of single-cell RNA sequencing in research on the inflammatory tumor microenvironment (figure was created with biorender.com accessed on 26 January 2023).

**Table 1 biomolecules-13-00344-t001:** **A** schematic overview of five single-cell RNA sequencing (scRNA-seq) methods.

Protocol	Single-Cell Capture	mRNA Reverse Transcription	cDNA Amplification	Library Construction	Reference
Tang method	Micromanipulation	poly(A) tailing + second-strand synthesis	PCR	Full-length	[11]
CEL-seq	LCM/Flow cytometry	second-strand synthesis	IVT	3′-Only	[12]
SMART-seq	Micromanipulation/LCM/Flow cytometry	template-switching method	PCR	Full-length	[13]
STRT-seq	LCM	template-switching method	PCR	5′-Only or 3′-Only	[14]
Drop-seq	Microfluidics	template-switching method	PCR	3′-Only	[15]

Abbreviations: CEL-seq: cell expression by linear amplification and sequencing; SMART-seq: switching mechanism at the 5′ end of the RNA transcript; STRT-seq: single-cell tagged reverse transcription sequencing; PCR: polymerase chain reaction; LCM: laser capture microdissection; IVT: in vitro transcription.

**Table 2 biomolecules-13-00344-t002:** A summary of the application of single-cell RNA sequencing in the inflammatory tumor microenvironment.

Cancer	Sample	Application					Combined with ST	Ref.
		Composition ^1^	Heterogeneity ^2^	Trajectory ^3^	Prognosis ^4^	Interaction ^5^		
Bladder Carcinoma (BC)	tumor samples & para tumor samples	Y	Y	Y	Y	Y		[54]
Gastric Cancer (GC)	tumor samples & adjacent mucosal samples	Y	Y	Y	Y	Y		[67]
Gastric Cancer (GC)	tumour samples & non-tumour samples	Y	Y	Y		Y		[76]
Multiple Myeloma (MM)	tumour samples & non-tumour samples	Y		Y		Y		[68]
Lung Adenocarcinoma (LUAD)	stage-I/II LUAD samples harboring EGFR mutations samples & tumor-adjacent Lung tissues	Y	Y	Y		Y		[69]
Acute Myeloid Leukemia (AML)	a novel animal model carrying a recurrent TET2 missense mutation	Y			Y	Y		[71]
Hepatocellular Carcinoma (HCC)	tumor samples & adjacent normal samples	Y		Y	Y	Y		[72]
Hepatocellular Carcinoma (HCC)	samples from primary or early-relapse HCC patients	Y	Y	Y	Y	Y		[73]
Gallbladder carcinoma (GBC)	chronic cholecystitis samples & treatment-naive GBCs samples & metastases samples	Y	Y		Y	Y		[74]
Esophageal Squamous Cell Carcinoma (ESCC)	tumor samples	Y	Y	Y	Y	Y	Y	[96]
Pancreatic Ductal Adenocarcinomas (PDAC)	tumor samples	Y	Y			Y	Y	[97]
Colorectal Cancer (CRC)	tumor samples & adjacent normal samples	Y	Y	Y	Y	Y	Y	[98]

^1^. Analyzing the composition; ^2^. revealing the heterogeneity; ^3^. describing the developmental trajectory; ^4^. predicting the prognosis; ^5^. revealing the interaction. ST: spatial transcriptomics; Ref: reference; Y: yes.

## Data Availability

Not applicable.

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
