# Peer review of "The Application of Single-Cell RNA Sequencing in the Inflammatory Tumor Microenvironment"

_biomolecules, 2023, doi:10.3390/biom13020344_

Round 1
Reviewer 1 Report
The manuscript entitled ‘’Progress on the application of single-cell RNA sequencing in the inflammatory tumor microenvironment” tries to descript single-cell RNA-sequencing applications in TME. The authors provide considerable information and literature more studies but the review manuscript needs more than a literature review. I have some concerns:
1. Authors should provide tables for the application scRNA-Seq in TME which show a summary and comparison. In a review paper tables and figures are very suitable for better understanding.
2. Figures: Authors should provide some figures that help to read for better understanding. Please design figures about the application scRNA-Seq in TME which will be special for your manuscript not similar to a previous study.
3. Replaced scRNA-Seq to scRNA-seq
Author Response
Thanks for the reviewer's comment, point by point response please find the attachment.

Reviewer 2 Report
The manuscript reviews the researches in the inflammatory tumor microenvironment base on scRNA-seq data analysis. The review is comprehensive and well written. It also includes reviewing the application of the emerging spatial transcriptomics,which is becoming hot spots. The following are my comments:
1. There exist some typos in the manuscript,such as line 143 'maker genes',line 177 's scRNA-seq'.
2. There should be a reference for the example of ESCC in lines 547-550.
3. In section 3.4, the authors use example of COVID-19 and pSS to illustrate scRNA-seq is helpful to uncover immune mechanisms. For the reivew should focus on the TME, I suggest the authos replace these diseases by tumor.
Author Response

(The authors gave the same response as above.)

Reviewer 3 Report
This review is a beautiful primer on single-cell technologies and the application of these technologies to tumor immunology to better understand the immune composition and function in tumorigenesis. Only comment is potentially reducing some of the references to physiology that are not related to the tumor field (e.g. COVID19, extensive explanation of roles of specific immune cells); reducing some of these paragraphs or integration into rest of the manuscript would help reduce length and provide a more cohesive reading experience.
Author Response

(The authors gave the same response as above.)

Reviewer 4 Report
The authors have reviewed the use of single-cell RNA sequencing in the study of the inflammatory tumour microenvironment (TME). The review covers the basic principles of single-cell RNA sequencing and immunology, and does an excellent job of taking the reader form this broad overview to the specific application of scRNA-seq in the TME. The work is of high quality and highly relevant.
Comments:
The manuscript has English language and editing issues and requires significant editing.
Author Response

(The authors gave the same response as above.)

Round 2
Reviewer 1 Report
I think the authors did all comments and the paper can be accepted